# Machine learning the electric field response of condensed phase systems using perturbed neural network potentials

Kit Joll [1], Philipp Schienbein [1,2] ✉, Kevin M. Rosso [3] &
Jochen Blumberger [1] ✉

The interaction of condensed phase systems with external electric fields is of major importance in a myriad of processes in nature and technology, ranging from the field-directed motion of cells (galvanotaxis), to geochemistry and the formation of ice phases on planets, to field-directed chemical catalysis and energy storage and conversion systems including supercapacitors, batteries and solar cells. Molecular simulation in the presence of electric fields would give important atomistic insight into these processes but applications of the most accurate methods such as ab-initio molecular dynamics (AIMD) are limited in scope by their computational expense. Here we introduce Perturbed Neural Network Potential Molecular Dynamics (PNNP MD) to push back the accessible time and length scales of such simulations. We demonstrate that important dielectric properties of liquid water including the field-induced relaxation dynamics, the dielectric constant and the field-dependent IR spectrum can be machine learned up to surprisingly high field strengths of about 0.2 V Å$^{-1}$ without loss in accuracy when compared to ab-initio molecular dynamics. This is remarkable because, in contrast to most previous approaches, the two neural networks on which PNNP MD is based are exclusively trained on molecular configurations sampled from zero-field MD simulations, demonstrating that the networks not only interpolate but also reliably extrapolate the field response. PNNP MD is based on rigorous theory yet it is simple, general, modular, and systematically improvable allowing us to obtain atomistic insight into the interaction of a wide range of condensed phase systems with external electric fields.

Electric fields are omnipresent in nature and technology. Their presence guides bumblebees to find nectar[1], and they are believed to cause a superionic ice VII phase on Venus[2]. They play a central role in a myriad of electronic and energy conversion devices, including field effect transistors, (super-)capacitors, batteries and solar cells. In chemistry, electric fields can be used to steer selectivities in catalysis[3]

and to control reactivities[4], whilst, in physics, they are used to accelerate particles close to the speed of light. The field strengths in these examples span an extraordinarily large range. Atmospheric fields in fair weather are on the order of $10^{-6}$ V Å$^{-1}$[5], and electric fields acting as floral cues can be as large as $10^{-5}$ V Å$^{-1}$[1]. In electrical components, the field strength can vary significantly depending on the actual design and

[1]Department of Physics and Astronomy and Thomas Young Centre, University College London, London, UK. [2]Department of Physics, Imperial College London, South Kensington, London, UK. [3]Pacific Northwest National Laboratory, Richland, Washington, UK. ✉e-mail: p.schienbein@ucl.ac.uk;
j.blumberger@ucl.ac.uk

the envisaged application from about $10^{-10}$ to $10^{-3}$ V $Å^{-1}$. Particle accelerators typically operate at a field strength of up to $5 \times 10^{-3}$ V $Å^{-1}$. At charged electrodes, field strengths on the order of 0.1 V $Å^{-1}$ [6] can be found, which also marks the onset of chemical bond activation[7] and electrofreezing of water[8,9]. The making and breaking of chemical bonds may require more than 1 V $Å^{-1}$[10,11].

Molecular dynamics (MD) or Monte Carlo simulations are the most suitable methods to understand and predict the interaction of liquids, electrolytes and liquid/solid interfaces with external electric fields because they sample the correct equilibrium distribution of molecular configurations. Many finite electric field simulations[12] have been carried out with classical force fields[13–15], which typically give a good description for pure solvents but tend to perform poorly on more complex systems where charge transfer and polarisation effects become important, e.g., electrode/electrolyte interfaces. Ab initio MD (AIMD) simulations[16] solve the electronic structure of the system from first principles at every MD time step (usually at the density functional theory (DFT) level of theory) and thus give the most accurate description for such cases. Indeed, AIMD simulations with finite electric fields have been carried out on a number of systems[17], ranging from crystals[12] and pure liquid water[18,19] to electrode/electrolyte interfaces[13,20–22]. A serious disadvantage of AIMD simulations is that they are still computationally demanding, with simulation times typically limited to a few 10 picoseconds and system sizes limited to a few 100 atoms depending on the density functional chosen. As such, AIMD simulations of important field-induced phenomena, such as dielectric relaxation, ionic conductivity[15], electric double layer formation[23] or capacitive charging, are prohibitively expensive due to the large number of atoms required to faithfully model such processes.

The advent of machine learning (ML) has transformed the field of MD simulations. It is now possible to carry out nanosecond MLMD simulations at virtually no loss in accuracy compared to AIMD by training ML potentials from just a few hundred explicit electronic structure calculations[24–26]. Originally, these ML models were designed to calculate potential energies and forces only, including schemes that decouple the total energy of the system into internal and environmental contributions[27,28]. Over the last years, a full bouquet of ML models has been published to predict dipole moments and polarizabilities[29–36], as well as other response properties, including atomic polar tensors (APT)[37]. Following these developments, some ML models were recently introduced that explicitly describe the interaction of a molecular system with external electric fields and were successfully applied to such systems and liquids[38–42]. In most of these approaches, the field dependence of the potential energy surface is explicitly or implicitly part of the ML potential[38–40,42]. Hence, the electric field is an input parameter to the ML potential, and therefore, training data for different field strengths are required, which is computationally expensive at the AIMD level.

Herein, we introduce a simple and robust alternative for MLMD with electric fields that learns the field response exclusively from zero-field molecular configurations, i.e., it circumvents the need to generate training configurations by running AIMD simulations with electric fields. We start from a standard potential energy surface and account for the interaction with the electric field in a perturbative manner by a series expansion truncated at first order, noting that the first order force term can be written in terms of the APT[37,43]. Two ML models are trained: one standard ML potential for the unperturbed potential energy surface (here, a committee neural network potential (c-NNP)[24,26,44]) and one ML model for the APT (here, an equivariant graph neural network denoted "APTNN"[37]). Combined, they form a "perturbed ML potential" (here, "perturbed neural network potential" (PNNP)). The two networks have no knowledge of electric field effects (neither explicitly nor implicitly) because they are both trained exclusively on configurations sampled at zero field, and the electric field itself is not an input of the machine learning models. The

interaction with the field during MD simulation is, therefore, entirely due to the first-order term in the series expansion given by the product of the APT represented by the graph neural network and the external field. The use of the APT and its ML representation to calculate the field-induced forces on the atoms has not been explored before, to our best knowledge, and represents the major conceptual advance of this paper.

PNNPs are based on rigorous physical principles and follow precisely the treatment of electric fields in quantum chemical calculations, where the total energy is expanded at zero field[45]. The accuracy can thus be systematically improved by adding higher-order contributions, such as polarizabilities and hyperpolarizabilities, in ML representation. Since we do not introduce or modify ML models, the advantages and accuracies, but also the limitations of the two employed ML models are inherited. Importantly, a PNNP follows the spirit of the modern theory of polarization[46] because the APT relates to a change of polarization and is thus not affected by the multivaluedness of the polarization in periodic boundary conditions. This makes PNNP applicable to a broad range of condensed phase systems typically modelled under periodic boundary conditions, including solids, liquids, and interfaces.

In the following section, we describe the PNNP approach in detail. After validation we apply the method to simulate the dielectric response of pure liquid water. We demonstrate reversible polarization and depolarisation of liquid water as the field strength is stepped up and down. Then the dielectric relaxation dynamics are analysed in detail, followed by the calculation of the dielectric constant from the response of polarization with respect to the field strength resulting in excellent agreement with the experimental value. Moreover, we show that PNNP correctly predicts the field-induced red shift in the O-H stretching mode and the field-induced blue shift in the librational mode of liquid water, in very good agreement with results from AIMD. In the Discussion section, we compare the PNNP to previously introduced ML methods for the calculation of molecular systems with electric fields and discuss current limitations and ways to overcome them.

## Results
### Perturbed Neural Network Potential (PNNP)
In our approach, the interaction of the atomistic system with a homogeneous external electric field $\mathbf{E}$ is treated perturbatively via a series expansion truncated at first order in the field[13,15,20,45]

$$\mathcal{H}_\mathbf{E}(\mathbf{r}^N, \mathbf{p}^N) = \mathcal{H}_0(\mathbf{r}^N, \mathbf{p}^N) - \mathbf{E} \cdot \mathbf{M}(\mathbf{r}^N), \qquad (1)$$

where $\mathcal{H}_0(\mathbf{r}^N, \mathbf{p}^N)$ is the total unperturbed Hamiltonian comprised of the kinetic energy of the $N$ nuclei with momenta $\mathbf{p}^N$ and the electronic potential energy depending on all nuclear positions $\mathbf{r}^N$, $\mathcal{H}_0(\mathbf{r}^N, \mathbf{p}^N) = E_{\text{kin}}(\mathbf{p}^N) + E_{\text{pot}}(\mathbf{r}^N)$, and $-\mathbf{E} \cdot \mathbf{M}(\mathbf{r}^N)$ is the perturbation induced by the electric field $\mathbf{E}$ acting on the total dipole moment of the system at zero field $\mathbf{M}(\mathbf{r}^N)$. The truncation to first order in the field is expected to be accurate for the weak and medium strong electric fields investigated in this work, as will be demonstrated further below. The field dependence of the dipole moment or, equivalently, higher order terms (polarizability, hyperpolarizability) may be added at stronger fields to account for the field-dependent perturbation of the electronic structure[47]. Applying Hamilton's equation of motion, we get the force acting on atom $i$

$$F_{i\xi} = -\frac{\partial E_{\text{pot}}(\mathbf{r}^N)}{\partial r_{i\xi}} + \sum_\zeta \frac{\partial M_\zeta}{\partial r_{i\xi}} E_\zeta, \qquad (2)$$

where $\zeta = x, y, z$ and $\xi = x, y, z$ represent the three Cartesian coordinates and $E_\zeta$ is the $\zeta$-component of $\mathbf{E}$. The first term is the force on the nuclei in the absence of an electric field, and the second term is the field-induced contribution which can be written in terms of the

transpose of the APT of atom $i$, $\mathcal{P}_i$[37,43] (not to be confused with the polarization **P**, Eq. (6)), with elements

$$\frac{\partial M_\zeta}{\partial r_{i\xi}} \equiv [\mathcal{P}_i^{\mathrm{T}}]_{\xi\zeta}. \tag{3}$$

In our approach, we train two ML models, one for the potential energy ($E_{\mathrm{pot}}(\mathbf{r}^N)$) and one for the APT ($\mathcal{P}_i$) and use the corresponding forces, Eq. (2), to carry out MD simulations in the presence of an external electric field. We emphasise that both quantities are trained without an external field present, in contrast to the schemes suggested before[38–40,42]. We use a committee[44] of 2nd generation high-dimensional Neural Network potentials[24,26] (c-NNP) to model $E_{\mathrm{pot}}(\mathbf{r}^N)$ and an E(3)-equivariant graph neural network to model the APT (APTNN) as recently introduced by one of us[37]. For details of the force implementation, we refer to the Methods section. In the following, the combined c-NNP and APTNN model for the electronic potential energy, including the field term, $E_{\mathrm{pot}}(\mathbf{r}^N) - \mathbf{E} \cdot \mathbf{M}(\mathbf{r}^N)$, is simply referred to as "perturbed neural network potential" (PNNP).

PNNP MD simulations give access to a number of important dielectric properties of solids, liquids and ionic solutions. The time derivative of the total dipole moment can be obtained by summing all APTs multiplied by the respective velocities of the nuclei, $v_{i\xi}$[37]

$$\dot{\mathbf{M}} \equiv \frac{\mathrm{d}\,\mathbf{M}}{\mathrm{d}t} = \sum_{i,\xi} \frac{\partial \mathbf{M}}{\partial r_{i\xi}} v_{i\xi}. \tag{4}$$

Thus, field-dependent IR spectra can be readily obtained from the autocorrelation function of $\dot{\mathbf{M}}$ sampled along PNNP trajectories (see Eq. (9) below). Time-integration gives the total dipole moment of the cell,

$$\mathbf{M}(t) = \mathbf{M}(t_0) + \int_{t_0}^{t} \mathrm{d}\,t' \dot{\mathbf{M}}(t'), \tag{5}$$

and the polarization **P**,

$$\mathbf{P}(\mathbf{t}) = \frac{\mathbf{M}(\mathbf{t})}{V}, \tag{6}$$

where $V$ is the volume of the simulation box. In passing, we refer to the modern theory of polarization in solids[47], where $\dot{\mathbf{M}}(t)/V$ is the transient current density that, when integrated over time, gives the itinerant dipole moment. The polarization in Eq. (6) is of major importance, allowing for the calculation of relevant dielectric properties including the dielectric constant (or relative permittivity, see Eq. (7) below), capacitance and ionic conductivity.

## Validation of PNNP

We demonstrate our approach by simulation of pure liquid water at room temperature for different electric field strengths ranging from about 0.002 to 0.2 V Å$^{-1}$. Details on the training of the c-NNP and APTNN for liquid water are given in the Methods section. Initialising a trajectory from an equilibrated water configuration at zero centre of mass momentum and running PNNP MD at an intermediate field strength of 0.0129 V Å$^{-1}$, we obtain a drift in the conserved total energy of $-2.2 \times 10^{-9}$ Hartree atom$^{-1}$ ps$^{-1}$ and a mean magnitude of the conserved centre of mass momentum of $8.4 \times 10^{-9}$ au, see Supplementary Fig. S1 A and B, respectively. Similar values are obtained for simulations at all other applied electric field strengths, and they are also typical for simulations without a field, attesting to the correct force implementation.

Next, we assess the quality of the PNNP by validating the predicted forces along the trajectories against an unseen test set. At each field

strength, 101 configurations are equidistantly extracted from 100 ps PNNP trajectories. Then the total force on each atom is computed using DFT reference calculations at the same field strength and compared to the total force from PNNP (data shown in Fig. 1A for an exemplary field strength of 0.0129 V Å$^{-1}$). We then decompose the DFT reference force into unperturbed and electric field-induced contributions by performing an additional DFT calculation on the same configurations but without the external electric field. The difference between the forces with and without the field isolates the field-induced forces. This allows us to separately validate the unperturbed c-NNP force contribution, first term on the right-hand side of Eq. (2), with the unperturbed DFT contribution (Fig. 1B) and the electric field-induced APTNN contribution, second term on the right-hand side of Eq. (2), with the field-induced DFT contribution (Fig. 1C). Evidently, we obtain a strong correlation between PNNP and DFT for the total forces and for the unperturbed and field-induced contributions.

The root-mean-square-error (RMSE) in the atomic forces as a function of field strength is shown in Fig. 1D. The total force RMSEs are on the order of 90 meV Å$^{-1}$ and thus in line with previously published high-dimensional NNPs and c-NNPs on various systems[44,48–50], yet somewhat larger than most recently reported values (see Discussion). Remarkably, we find that the RMSE for the unperturbed contribution remains almost constant at about 90 meV Å$^{-1}$ even at field strengths up to 0.2057 V Å$^{-1}$, although the training set is only composed of unperturbed, i.e., zero-field equilibrium configurations of liquid water. The field-induced force contributions are about two orders of magnitude smaller than the unperturbed contribution, and so are their RMSEs. Notably, the field-induced force RMSE increases in proportion with the field strength from $8.6 \times 10^{-2}$ meV Å$^{-1}$ at 0.0026 V Å$^{-1}$ to 9.05 meV Å$^{-1}$ at the largest simulated field of 0.2057 V Å$^{-1}$. Dividing the RMSE by the range of field-induced forces one obtains 1.1% error at 0.0129 V Å$^{-1}$ and 1.2% error at 0.2057 V Å$^{-1}$. Alternatively, dividing the RMSE by the root-mean-square of the reference DFT forces, as is commonly done in the literature[50] (see also Supplementary Note 4), we obtain a relative RMSE for the field-induced contribution of $\approx 8.2\%$ up to a field strength of about 0.02 V Å$^{-1}$ and 9.8% at 0.2057 V Å$^{-1}$ (Fig. 1E). Hence, the relative force error on both error metrics exhibits remarkably constancy across the range of field strengths investigated.

Lastly, we validate the calculation of the total dipole moment along the PNNP MD trajectories as obtained by integration of the time derivative of the dipole moment according to Eq. (5). We find that the dipole moment tracks the reference dipole moments obtained from explicit DFT calculations very well for about 10–100 ps depending on the field strength (see Supplementary Fig. S1C). However, at longer times, deviations become larger due to the accumulation of errors when integrating over the finite time steps, even though $\dot{\mathbf{M}}(t)$ is accurately reproduced. This problem is addressed here by calculating reference DFT dipole moments along PNNP MD trajectories in periodic intervals and integrating the time derivative of the dipole moment obtained from APTNN only from one DFT reference value to the next. Using this integration procedure, we calculated the mean dipole moment averaged over the 100 ps PNNP MD trajectories, $\langle \mathbf{M}(t) \rangle$, as a function of the time interval between two DFT reference values. The relative errors with respect to the reference DFT mean dipole moments averaged over the same 100 ps PNNP MD trajectories (calculated using a sampling frequency of 1 ps$^{-1}$) are shown in Fig. 1F. We find that the error decreases rapidly, to below 5% for spacing between two DFT reference dipole moments of 10 ps, for all field strengths. This integration procedure with a spacing of 10 ps is used for the total dipole moments reported in this work. Hence, in practice, only a very small number of additional DFT calculations are necessary to accurately calculate the dipole moment along the PNNP MD trajectories at a resolution that is only limited by the MD time step.

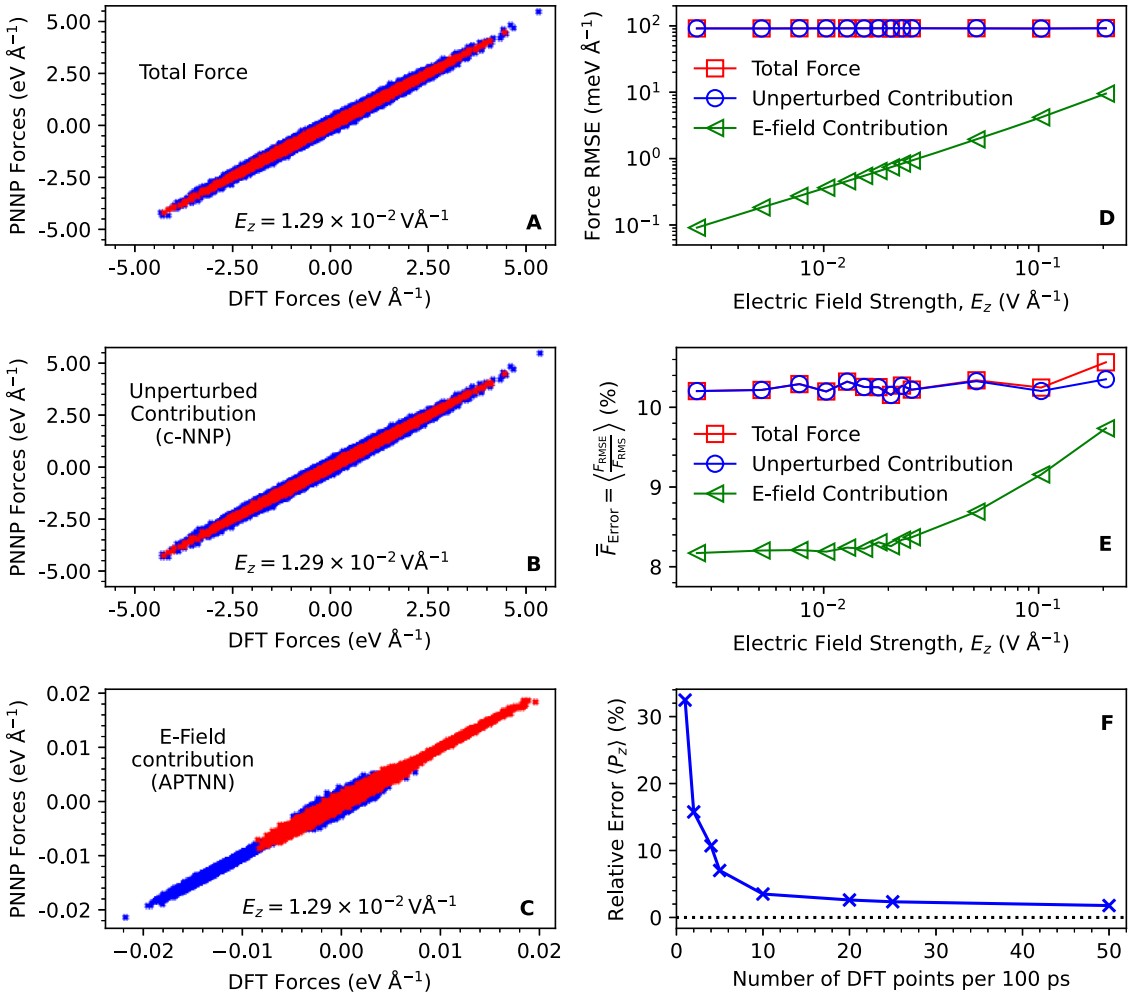

**Fig. 1 | Error metrics of the trained Perturbed Neural Network Potential (PNNP).** Scatter plots are presented comparing the predicted PNNP forces on the atoms with the corresponding reference density functional theory (DFT) forces for the total force (**A**), which is broken down into the unperturbed force contribution predicted by the committee Neural Network Potential (c-NNP) (**B**) and the field-induced force contribution predicted by the Atomic Polar Tensor Neural Network (APTNN) (**C**). The field strength was 0.0129 V Å$^{-1}$, and 101 configurations were tested against. Forces acting on O and H atoms are colour-coded in blue and red, respectively. The root-mean-square-error (RMSE) of the PNNP forces relative to DFT forces as a function of the field strength is shown in panel (**D**), where the total force, unperturbed force contribution and field-induced force contribution are depicted in squares, circles, and left-facing triangles, respectively. The corresponding relative RMSE, defined by the ratio of the force RMSE ($F_{RMSE}$) to the root mean square of the DFT forces ($F_{RMS}$), is displayed as a percentage in panel (**E**) (see Supplementary Note 4 for detailed definition). The relative error of the mean polarization in the direction of the field ($\langle P_z \rangle$), obtained by integrating the total dipole moment time derivative (Eq. (5)), is shown in panel (**F**) as a function of the number of DFT reference calculations of dipole moments per 100 ps, see main text for details. Source data are provided in the Source Data files.

## Orientational relaxation dynamics

Having validated the forces and polarization against DFT reference data, we now apply the PNNP to simulate the relaxation of water orientation in response to interaction with an electric field. We extract 20 independent configurations from a sample of liquid water equilibrated for 1 ns with field-free c-NNP MD and use them as starting configurations for 40 ps PNNP MD simulations, each run at a field strength of 0.0257 V Å$^{-1}$. In addition, we take the same initial configurations and perform explicit AIMD simulations at the same field strength and integration time step (1 fs). The orientation of the water molecules at a given time, $t$, is described by the angle between the direction of the applied electric field and the bisector between the two intramolecular OH bond vectors averaged over all water molecules, $\langle \Theta \rangle(t)$. The results are shown in Fig. 2. At $t = 0$, the initial average orientation is very close to 90° corresponding to the expectation value of the randomly oriented water molecules at equilibrium. When the field is switched on at $t = 0$, the relaxation dynamics obtained from PNNP MD are in very good agreement with the results from AIMD. Exponential fits give time constants $\tau = 5.9$ ps$^{-1}$ ($R^2 = 0.99$) for PNNP

compared to $\tau = 6.6$ ps$^{-1}$ ($R^2 = 0.99$) for AIMD. Moreover, the PNNP and AIMD simulations converge to nearly the same final average angle of 60° and 61°, respectively. Since the standard deviations in $\langle \Theta \rangle$ overlap for the two methods (shaded areas in Fig. 2), we ascribe the remaining small differences to statistical uncertainty. A much larger number of trajectories totalling several nanoseconds would be needed to clarify this point, but this is unfeasible due to the computational expense of AIMD.

## Electric field sweep

In the following we vary the strength of the field in time and monitor the structural and dielectric response of the water sample as obtained from PNNP simulations. The results are presented in Fig. 3. The temporal profile of the applied electric field strength is shown in panel A, the average water orientation $\langle \Theta \rangle(t)$ in panel B and the total dipole moment in the direction of the applied electric field, $M_z(t)$, in panel C. We obtain an approximately linear response in the average orientation and a concomitant linear increase in the dipole moment as the field is stepped up from 0 to 0.0154 V Å$^{-1}$ (in increments of 0.0026 V Å$^{-1}$). This

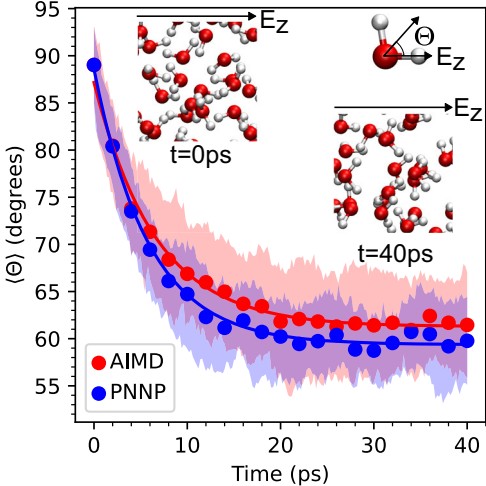

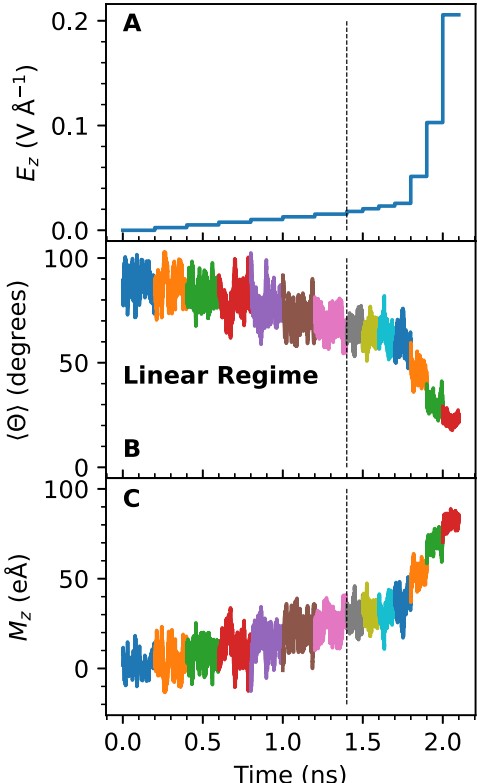

**Fig. 2 | Electric field-induced orientational relaxation of liquid water.** The change in the average orientation of the water molecules defined by $\langle \Theta \rangle$ is shown for the Perturbed Neural Network Potential molecular dynamics (PNNP MD) (blue circles), and ab initio molecular dynamics (AIMD) (red circles) after an electric field along the z-direction with field strength $E_z = 0.0257$ V Å$^{-1}$ is switched on at $t = 0$. The angle $\Theta$ is defined for each water molecule as the angle between the bisector of the two intramolecular OH bond vectors and the applied electric field vector, as illustrated in the inset at the top right corner. The average, $\langle \Theta \rangle$, is obtained by averaging $\Theta$ overall water molecules in a given configuration. The shaded areas indicate the standard deviations obtained from 20 independent trajectories. The data are fit to exponential decay functions (solid lines). The insets show a snapshot of an equilibrated water sample at $t = 0$, where $\langle \Theta \rangle \approx 90°$ corresponds to randomly orientated water molecules, and a snapshot at $t = 40$ ps, where the water molecules are polarised at an average angle of about 60°. Source data are provided in the Source Data files.

**Fig. 3 | Electric field sweep for liquid water.** The applied electric field along the z-direction, $E_z$, the average angle $\langle \Theta \rangle$ measuring the orientation of water dipoles along the z-axis (as defined in Fig. 2) and the dipole moment of the simulation cell along the z-direction, $M_z$, are shown in panels (**A**–**C**), respectively, as a function of the simulation time. The vertical dashed line indicates the upper limit of the linear response regime at 0.0154 V Å$^{-1}$. It was determined from the data shown in Fig. 4 as the threshold above which the polarization versus electric field response starts to deviate from linearity. Perturbed Neural Network Potential molecular dynamics (PNNP MD) simulations were carried out for 200 ps at each field strength until the end of the linear regime was reached and for 100 ps at each field strength in the non-linear regime. A change in colour indicates a step change in the electric field strength. $M_z$ was obtained by integration of the Atomic Polar Tensor Neural Network (APTNN) prediction of the time derivative of the dipole moment according to Eq. (5), as explained in the main text. Source data are provided in the Source Data files.

is in line, albeit somewhat lower than, previously published estimates for the upper bound of the linear regime, 0.03 to 0.07 V Å$^{-1}$ [17,51]. At larger field strengths, the dielectric response becomes weaker, indicating that the non-linear regime is reached. At 0.2057 V Å$^{-1}$, a strong orientational alignment along the field direction is observed, $\langle \Theta \rangle = 20°$. Notice that at these high field strengths, the PNNP still predicts reasonably accurate forces (see Fig. 1D, E) despite the absence of electronic polarization terms and any field-dependent training data. At 0.4114 V Å$^{-1}$, we observe water splitting into a proton and a hydroxide ion, in line with previous reports that chemical bond activation occurs at these field strengths [7]. Here, one could train the c-NNP further to include proton and hydroxide species, and thereby test even larger field strengths. However, an accurate description of these species would require explicit inclusion of nuclear quantum effects [52,53], which is beyond the scope of this work. Instead, an additional set of simulations is run where the field strength is reversed from the value at the end of the linear regime, 0.0154 V Å$^{-1}$, down to 0 in increments of 0.0026 V Å$^{-1}$. That sweep is detailed in Supplementary Fig. S2. We obtained very similar mean orientations and dipole moment as for the forward sweep, demonstrating that the sample can be reversibly polarized and depolarised.

## Dielectric constant

The time series of dipole moments shown in Fig. 3C are time-averaged for each applied electric field strength to obtain the mean of the polarization (Eq. (6)) along the field direction, $\langle P_z \rangle$, as a function of the field strength. The results are shown in Fig. 4 for the forward and backward sweeps (data in blue and red, respectively). The data points in the linear regime are shown magnified in the inset of Fig. 4. They are fit to straight lines, and the slopes used to obtain the static dielectric

constant, $\epsilon_r$, according to Eq. (7),

$$\epsilon_r = 1 + \frac{1}{\epsilon_0} \frac{\partial \langle P_z \rangle}{\partial E_z}, \qquad (7)$$

where $\epsilon_0$ is the vacuum permittivity. We obtain values $\epsilon_r = 77.9 \pm 2.7$ ($R^2 = 0.99$), $80.6 \pm 2.7$ ($R^2 = 0.96$) for forward and backward sweep and $\epsilon_r = 79.3 \pm 2.2$ ($R^2 = 0.99$, data in green) from a linear fit to the weighted average data for forward and backward sweep, in agreement with the experimental value of $\epsilon_r = 78.4$ [54]. Similar values would be obtained using only one data point due to the robust linear correlation. Remarkably, the dielectric constant was converged after only about 175ps of simulation time per applied electric field (see Supplementary Fig. S3), similar to what was previously reported for classical MD simulations employing finite field Hamiltonians [13]. This is an order of magnitude less simulation time than what is required to calculate the dielectric constant from the fluctuations of the polarization [13,55–57],

$$\epsilon_r = \epsilon_\infty + \frac{1}{3\epsilon_0 k_B V T} \left( \langle \mathbf{M}^2 \rangle - \langle \mathbf{M} \rangle^2 \right), \qquad (8)$$

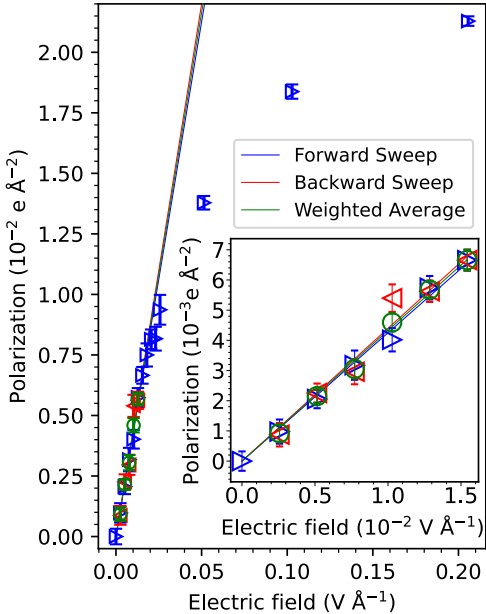

**Fig. 4 | Polarization and dielectric constant.** The time-averaged polarization of liquid water, $\langle P_z \rangle$, is shown as a function of the applied electric field along the $z$-direction, $E_z$. Data points obtained from the field sweep in the forward (backward) direction, where the field is stepped up (down), are depicted in blue (red) triangles. Data points in the linear response regime between 0 and 0.0154 V Å$^{-1}$ are fit to straight lines with zero intercept. The data were averaged over the Perturbed Neural Network Potential molecular dynamics (PNNP MD) trajectories shown in Fig. 3 (Supplementary Fig. S2) for forward (backwards) sweep and converted to polarization according to Eq. (6). The number of polarization samples averaged over was 180001 in the linear regime and 80001 in the non-linear regime. The standard error of the mean (SEM) is displayed and was calculated as the standard deviation of the sample divided by the square root of the sample size, adjusted for statistical inefficiency. To this end, a literature autocorrelation time of 10ps was used[99]. The weighted averages over data from forward and backward sweeps are depicted in green circles and fit into a straight line. The linear response regime is shown enlarged in the inset. The slope is used to obtain the dielectric constant according to Eq. (7). Source data are provided in the Source Data files.

where the total dipole moment, **M**, is sampled at zero electric field and $\epsilon_\infty = 1.72$[58] is the optical dielectric constant for liquid water. Indeed, sampling the total dipole moment **M** along zero field c-NNP trajectories, we obtain a converged value $\epsilon_r = 90.6 \pm 1.7$ only after 3 ns (see Supplementary Fig. S4), similarly as in previous simulations[59]. The reason for the difference in the dielectric constant from the field sweep and from zero field simulation is not known but could be due to a number of reasons. The fluctuations of the polarization are likely to be more sensitive to simulation details than the mean values, e.g., thermostat used, finite system size and remaining inaccuracies of the PNNP. Moreover, the approximations made to derive the dielectric constant in terms of the polarization fluctuations from Eq. (7)[60] could also contribute to the difference.

### Field-dependent IR spectra

The APTs and nuclear velocities along the PNNP trajectories give direct access to the time derivative of the dipole moment, **Ṁ**, according to Eq. (4), and the frequency ($\omega$)-dependent Beer-Lambert absorption coefficient of IR spectroscopy, $\alpha(\omega)$,

$$\alpha(\omega)n(\omega) = \frac{\pi}{3Vc\epsilon_0 k_B T} \frac{1}{2\pi} \int_{-\infty}^{\infty} dt\, e^{-i\omega t} \left\langle \dot{\mathbf{M}}(0)\dot{\mathbf{M}}(t) \right\rangle, \quad (9)$$

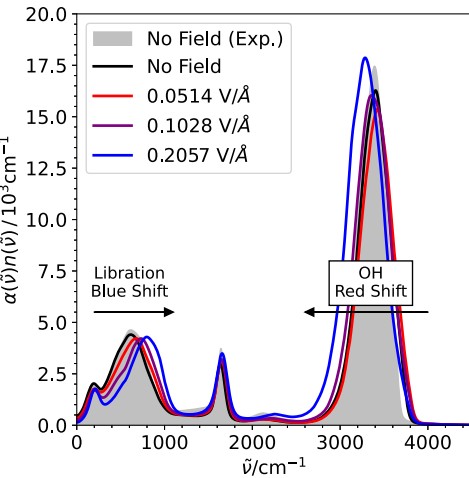

**Fig. 5 | Field-dependent Infrared (IR) spectrum of liquid water.** The product of Beer-Lambert absorption coefficient and refractive index, $\alpha(\tilde{\nu})n(\tilde{\nu})$, Eq. (9), is shown as a function of the vibrational wavenumber $\tilde{\nu}$ for different applied electric field strengths, as indicated. The time derivative of the polarization is directly obtained from the Atomic Polar Tensor Neural Network (APTNN) and the atomic velocities sampled along the Perturbed Neural Network Potential molecular dynamics (PNNP MD) trajectories via Eq. (4). The experimental spectrum at zero field (shown in shaded grey) is taken from ref. 61. Source data are provided in the Source Data files.

where $n(\omega)$ is the frequency-dependent refractive index, $c$ is the speed of light in vacuum, $k_B$ is the Boltzmann constant and $T$ the temperature. In Fig. 5 we present the calculated spectrum as obtained from c-NNP MD at zero field strength and from PNNP MD at finite field strengths. The experimental IR spectrum at zero-field[61] is very well reproduced using the RPBE-D3 functional, as reported previously[37]. The spectrum remains remarkably insensitive to the presence of an electric field in the linear regime and beyond, up to about 0.0514 V Å$^{-1}$. For larger fields, we observe a systematic red-shift of the intramolecular OH stretching vibration at around 3500 cm$^{-1}$, by approximately 100 cm$^{-1}$ at 0.2057 V Å$^{-1}$. A redshift of the OH stretch is generally ascribed to the formation of stronger intermolecular hydrogen bonds[62–64], here induced by the external electric field. This notion is further supported by the systematic blue shift of the librational band at around 600 cm$^{-1}$ by approximately 200 cm$^{-1}$ at 0.2057 V Å$^{-1}$ suggesting that the interaction with the field leads to a stiffening of the potential for rotational motion of the water molecules[65]. This is in line with a previous study describing water at these field strengths to be more ice-like[8,9]. The trends in the IR spectra agree very well with previously published results obtained from other approaches that differ in many aspects from our method, e.g., AIMD simulations with external electric fields[8] and MLMD simulations using the FIREANN-wF model[42]. The computed band shifts are thus a sensitive fingerprint of the local electric fields. In turn, such calculations may provide a way to estimate the local field strength in a sample from their measured IR spectrum.

## Discussion

In this work, we extend MLMD simulations to include the interaction with an external electric field by adding the field-induced perturbation up to first-order to the unperturbed Hamiltonian (Eq. (1)). The resulting force equation (Eq. (2)) allows us to separately calculate the unperturbed forces from a standard machine learning potential (here: a c-NNP) and the field-induced forces from the APTNN. The force calculation, therefore, rigorously follows the laws of electrostatics without any additional approximations.

The resulting scheme termed PNNP MD has several advantages: First, the approach is modular because the electric field contribution is

independent of the employed unperturbed potential energy surface. As such, the APTNN can be coupled with any ML potential (not only NNPs) or even with force fields to include the interaction with an external electric field. The level of theory of the reference electronic structure calculations for the training of APTNN and ML potential can be chosen differently in accord with the accuracy requirements for unperturbed and perturbed potentials. Moreover, the modular approach allows one to assess the accuracy of each component separately. Second, APTs are well-defined for any atomistic system, and thus, APTNNs can be trained out of the box, without requiring any conceptual tailoring or adjustment or the use of arbitrarily defined proxies (e.g., atomic charges, molecular dipole moments, or similar)[37]. Importantly, APTs are uniquely defined when using periodic boundary conditions as they quantify a change in the total dipole moment or polarization. This is in contrast to the total dipole moment, which is multi-valued in periodic boundary conditions[46]. Third, the approach does not require sampling training configurations as a function of the applied electric field. Yet, the interaction with electric fields is predicted at near DFT reference accuracy up to high field strengths on the order of 0.2 V Å$^{-1}$. Fourth, the method can be systematically improved by including higher order terms to the perturbation, e.g., polarizability and hyperpolarizability[45]. This would allow for more accurate simulations at very high field strengths exceeding 0.2 V Å$^{-1}$.

These features make our PNNP method distinct from other approaches introduced to model the interaction with external electric fields in ML simulations[38–40,42]. The FIREANN-wF model[42] is trained on atomistic forces in the presence of an electric field to perform finite-field simulations of liquid water and thus requires field-dependent training data. The model adds a pseudo atom to each real atom which is explicitly dependent on the electric field and thereby captures the response to an applied electric field in an effective atomic dipole moment. It has been used remarkably successfully to calculate response functions, such as dipole moments and polarizabilities, and subsequently, field-dependent vibrational spectra of liquid water were computed from these quantities. In principle, this approach allows one to accurately model arbitrarily large electric fields and field gradients, as the relevant polarized configurations are explicitly included in the training set. By contrast to PNNP, in FIREANN-wF the field-induced contribution to the total energy and forces is not learned explicitly but "implicitly in the force-only training"[42]. It remains to be seen whether implicit learning is a robust strategy that can be applied to a wide range of systems.

FieldSchNet[39] incorporates external field effects by introducing vector-valued representations of atomic environments, e.g., utilizing fictitious atomic dipole moments. It then uses vector fields to model interactions between molecules and arbitrary external environments by adding terms for dipole-field and dipole-dipole interactions. In contrast to PNNP, it is trained on data that include molecular structures in various dielectric environments and field strengths. Electrostatic response properties of the field are available as analytical derivatives of the network and have been used to compute vibrational spectra from MD simulations conducted at zero field. Certain limitations of this method have recently been pointed out, e.g., an incomplete description of the field-system interaction when atomic dipoles are orthogonal to the applied field direction[42].

Complementary to the above methods, SCFNN[40] relies on training the position of Wannier centres around water molecules. The method has been introduced with the main goal of incorporating long-range electrostatics but also allows one to evaluate configurations in the presence of external electric fields. The method can indeed be generalised such that molecules other than water can be treated, but "the set of possible molecules must be known in advance"[40]. Moreover, a reference frame around each kind of molecule needs to be defined; in the case of a water molecule, this was done by associating four Wannier centres with the molecule and examining their coordinates within the local molecular frame. This construction thus likely introduces an overhead when more molecules need to be treated, molecular frames cannot be constructed (e.g., a simple ion), or reactions occur during the MD simulations (e.g., proton transfer in water).

Returning to PNNP MD, we have shown that our method predicts accurate dipolar response dynamics (Fig. 2) and electric field-dependent IR spectra when compared to AIMD (Fig. 5) as well as an accurate dielectric constant when compared to the experimental value (Fig. 4). The almost quantitative agreement obtained for the dielectric constant is remarkable because one generally cannot expect perfect agreement between electronic structure calculations at the level of DFT and experiments. Moreover, nuclear quantum effects[52,53] were ignored in our simulations. The RPBE-D3 functional, in combination with classical MD simulation, is known to describe water, aqueous solutions and interfaces over a wide range of temperatures and pressures very well[37,59,65–70]. With the help of PNNP simulations, we could show that the strong performance of the RPBE-D3 functional in describing radial distribution functions, self-diffusion, orientational relaxation processes, and vibrational spectra of liquid water with respect to experimental data extends to its dielectric response properties. This is in part because the remaining deficiencies of this functional tend to be effectively compensated by missing nuclear quantum effects. Indeed, our computed IR spectrum at zero-field is in almost quantitative agreement with experimental data. Moreover, our computed peak shifts in the IR spectra in the presence of an applied electric field are in almost quantitative agreement with the results reported for the FIREANN-wF model[42]. Therein a more accurate and computationally expensive revPBE0-D3 hybrid functional was employed in combination with path integral MD simulations to explicitly account for nuclear quantum effects.

The performance of PNNP might be surprising considering that the typical field-induced contribution to the total force, in the order of 10 meV Å$^{-1}$ at moderate field strengths of 0.01 V Å$^{-1}$ (Fig. 1C), is even smaller than the RMSEs of the unperturbed force contribution and the total force, about 90 meV Å$^{-1}$ (Fig. 1D). We explain this by noting that the error in the unperturbed force contribution is approximately Gaussian distributed and does not have a directional preference whereas the field-induced force contribution has, of course, a net direction along the field. Thus, when averaged over many configurations and atoms, the error in the unperturbed force contribution cancels out, whereas the field-induced force contribution does not. Therefore, the RMSE in the unperturbed force contribution does not compromise the accuracy of the electric field response.

We would like to point out that the total force RMSE of our PNNP, about 90 meV Å$^{-1}$, is in line with typical values that have been previously reported for ML models, see e.g., refs. 26,48–50. Yet, some recent studies reported RMSEs that are about a factor of 2–3 smaller, for instance, an RMSE of 39.4 meV Å$^{-1}$ was reported for the FIREANN-wF model[42]. The training set used herein for c-NNP was not very exhaustive and consisted of only 260 configurations of a 128-meter molecule box. Hence, it is much smaller than the training set used in ref. 42. Similarly, the APTNN was trained on only 27 randomly selected configurations of liquid water[37]. Another important aspect is the quality of the underlying reference DFT calculations. For consistency reasons, we employed exactly the same DFT setup that was used in previous AIMD studies on liquid water[67,68]. In particular, we adopted a kinetic energy cutoff of 600 Ry, whereas, in the case of the FIREANN-wF model[42], a very tight cutoff of 1200 Ry was used. It is well known that tighter convergence of the electronic structure calculations leads to less noise and smaller RMSEs in the ML model[26]. Thus, we expect that the RMSEs of our ML models could be further lowered by increasing the training set in combination with the use of tighter convergence criteria for the DFT calculations. The strong performance of our PNNP model compared to AIMD and experimental data suggests that the current RMSE is sufficiently low for the purpose of this study.

As with any method, our current implementation of PNNP MD has some limitations, which we would like to discuss in the following. First, the itinerant cell dipole moment is obtained by integrating the current density according to Eq. (5). Therefore, cell dipole moments are only indirectly obtained by numerical integration, in contrast to previously published techniques[39,40,42]. To compensate for the accumulation of integration errors, the integration is restarted periodically by explicitly calculating the dipole moment by the underlying electronic structure method (Fig. 1F). While this significantly reduces the amount of electronic structure calculations required, the additional DFT calculations may become performance-limiting for very large system sizes that can no longer be routinely treated at DFT level (e.g., 1000s of water molecules). Yet, the DFT calculations do not restrict the accessible time scale that can be accessed by PNNP MD. As we show in Fig. 1F, it is sufficient to carry out DFT calculations in periodic intervals (10 ps) on the time scale accessible to PNNP MD simulations (tens of ns). Notably, these DFT calculations are only post-processing of the PNNP MD such that they can be calculated in parallel. Moreover, many relevant properties do not require knowledge of the itinerant cell dipole moment, e.g., vibrational spectra, or transport properties, where the current density (Eq. (4)) is sufficient. The latter is readily available in the PNNP scheme and does not require numerical integration.

While the APT rigorously accounts for all long-range electrostatics and non-local charge transfer effects, our neural network representation of the APT (APTNN) only uses local descriptors, i.e., it does not contain explicit long-range electrostatics information. This is similar in spirit to the 2nd generation c-NNP used here to represent the interatomic potential. ML models with local descriptors are bound to fail for vapour phases or apolar media where long-range electrostatics are no longer effectively screened, or in situations where the electric field induces a long-range charge transfer. Yet, this is where the modularity of our PNNP approach offers a distinct advantage because it allows us to replace the underlying ML models as required. For example, the 2nd generation c-NNP currently modelling the interatomic potential can be replaced by a 4th generation c-NNP[26,71,72] which explicitly takes long-range electrostatics and non-local charge transfer into account. Similarly, the APTNN with local descriptors could eventually be replaced with a version including non-local descriptors. The idea introduced here to perturb an interatomic potential with an electric field using the APT is generally valid. It is a matter of the underlying machine learning models to incorporate non-locality when needed.

We also note that the perturbation series expansion in the electric field Eq. (4) only contains the dipolar term. The low relative RMSE in the field-induced force contribution suggests that higher order terms (dipole polarizabilities and hyperpolarizabilities) are not needed up to high field strengths of about 0.2 V Å$^{-1}$ (Fig. 1E). Yet, the RMSE slowly but steadily increases at this point indicating that dipole polarizability becomes increasingly important in this electric field regime. In principle, the polarizability tensor can be learned in a similar way as the APT, and we are currently exploring efficient schemes for this purpose. Finally, another limitation of the current scheme is that we only considered homogeneous electric fields. For simulation under inhomogeneous fields induced, e.g., by an STM tip or an electrode interface, one would need to supplement the multipole expansion Eq. (4) with the quadrupolar interaction terms (since field gradients interact with quadrupole moments). This could be done using the Born charges obtained from the APT for the calculations of the quadrupole moment.

In conclusion, we have implemented an ML methodology, denoted PNNP, to run MD simulations in the presence of an external homogeneous electric field. Key to this development is the APT, which is trained by an ML model and used to compute the field induced forces. The latter are combined with the forces obtained from a standard ML potential describing the interatomic potential. The method is modular, it makes use of the APT which is well defined in periodic boundary conditions, it does not require training as a function of the applied electric field and it is systematically improvable.

We performed PNNP simulations at several different field strengths and compared our results against reference AIMD calculations and experimental literature data. In general, we found very good to excellent agreement for all properties investigated. We demonstrated reversible orientational polarization and depolarisation of the water molecules as the electric field is stepped up and down, with orientational relaxation times within the statistical accuracy of the AIMD data (Fig. 2). This permitted successful calculation of the relative permittivity, $\epsilon_r$, on simulation time scales that are an order of magnitude shorter than in more standard approaches that suffer from the very slow convergence of the polarization fluctuations at zero field (Fig. 4). The value obtained was in good agreement with experiment validating our approach in the linear response (low electric field) regime. We also calculated the IR spectra and compared them with previous explicit AIMD simulations[8]. The systematic peak shifts for libration and intramolecular OH stretch vibration at high electric fields match AIMD reference data very well, thereby validating our approach also for the non-linear response (high electric field) regime.

We expect PNNP MD to become a useful tool for the ab initio-level simulation of a wide range of condensed phase systems interacting with external electric fields and for the calculation of electric field-dependent properties not investigated in this work, including ionic conductivity and capacitance. The implementation of higher-order terms in the energy expansion with respect to the electric field will enable accurate simulations at even higher field strengths and give access to Raman and Sum-Frequency-Generation spectra of condensed phase systems within the APT framework. Moreover, owing to the modularity of our approach, one can readily take advantage of future developments in non-local ML potentials, which might be necessary for simulation in media where dielectric screening is not as strong as in water.

## Methods

### Implementation of PNNP

The unperturbed potential energy, $E_{pot}$, and the corresponding forces, the first term on the right-hand side of Eq. (2), are modelled by a committee of 2nd generation high-dimensional Neural Network potentials[24,26], as recently implemented in the `cp2k` software package[44]. The field-dependent force contribution related to the APT, the second term on the right-hand side of Eq. (2), is modelled by an APTNN, which is based on the E(3)-equivariant graph neural network `e3nn`[73] based on the `PyTorch` library[74]. A new force evaluation environment was added to `cp2k`, linking the Fortran-based `cp2k` and the Python/C++ based `PyTorch` in a client/server approach. Inspired by the i-PI implementation[75] `cp2k` launches and connects to a Python server that waits to receive configurations and sends back APTs predicted by the APTNN model. The APTs are then used in `cp2k`, to evaluate the corresponding force contributions. They are added to the c-NNP forces using the built-in `mixing` force environment to obtain the total forces for propagation of the atoms using the velocity-Verlet algorithm.

### Training of c-NNP and APTNN

The committee members of the c-NNP were trained using `n2p2`[76]. The parameters for the neural network were taken from a previous ML potential study on liquid water[59], in conjunction with generic symmetry functions[44]. The c-NNP is trained on the energies and forces obtained from DFT calculations at the level of RPBE-D3, as detailed further below using the active learning procedure reported in ref. 44. With regard to the APT, we use the recently published APTNN developed by one of us[37] containing APTs from only 27 snapshots of an equilibrated 128 water molecule box in its training set at the level of RPBE-D3[37]. The resultant 10,368 APTs were obtained by single-point

finite difference calculations, which required significant computational effort. Notice that the goal of the present work was to introduce the PNNP approach and to give a proof-of-principle demonstration, not to optimise the training protocol. Here, we just used the largest training set that was already available from our previous work[37]. This has allowed us to benchmark the accuracy of PNNP whilst ensuring that possible errors from using too small training sets are minimised.

We are very confident that the computational effort for the generation of training data for APTNN can be significantly reduced in the near future. First, we previously showed that the IR spectrum requires only 9 training configurations for a 128 water molecule box (i.e., 3456 APTs) to accurately reproduce the reference spectrum, reducing the required reference calculations by a factor of three. We anticipate that this applies to other observables of interest as well. Second, our current training strategy for the APTNN has not yet been optimised. Calculating APTs for all atoms in a given configuration likely includes redundant data due to the high correlation between APTs of neighbouring atoms. Training on a per-atom basis, where APTs are calculated for a subset of atoms only, is expected to be more efficient. This method could also be combined with active learning, where the network iteratively selects atoms for the training set; for example, by using a query-by-committee approach as in the c-NNP[44]. We expect that these two strategies allow us to reduce the number of reference APT calculations by an order of magnitude, if not more. Finally, the method for calculating APTs could also be made more efficient, for example, by using analytic density functional perturbation theory[77] rather than finite difference calculations. Exploring these strategies is beyond the scope of this manuscript but will be addressed in future work.

### Reference DFT calculations

Electronic structure calculations for the generation of reference training and test data were performed using `cp2k`[78] version 2023.1 and the quickstep module[79]. The RPBE functional was evaluated by the `libxc` package[80] and supplemented by D3 dispersion corrections[81]. We employed a mixed Gaussian orbital/plane wave (GPW) basis set[82] with a plane wave cutoff of 600 Ry and a relative cutoff of 20 Ry, and the Gaussian orbitals are constructed using the triple-$\zeta$ quality TZV2P basis set[83], including polarization functions as employed previously[65–68]. Core electrons are described by norm-conserving Goedecker-Teter-Hutter (GTH) pseudopotentials[84,85]. Homogeneous electric fields were treated by the approach introduced by Umari and Pasquarello[86] as implemented in `cp2k`. Dipole moments of the simulation cell ($\mathbf{M}$) were obtained from the DFT calculations via maximally localised Wannier functions[16].

### Charge conservation

Charge conservation is strictly enforced in PNNP via the acoustic sum rule for the atomic polar tensors[77,87,88],

$$\sum_{i}^{\text{atoms}} \frac{\partial M_\zeta}{\partial r_{i\xi}} = 0 \ \ \forall \ \zeta, \xi. \tag{10}$$

This is done by calculating the sum of every component divided by the number of atoms and distributing any small excess evenly across all atoms in the system. Note that this correction is akin to correcting atomic charges, as it has previously been done in the literature, distributing any excess charge evenly across the total system[89–91].

### Simulation details

All calculations, including training, testing and production runs, were carried out for a 128 water molecule box of length 15.6627 Å employing periodic boundary conditions (density 0.996 kg L$^{-1}$). Finite field PNNP MD simulations were carried out in the NVT ensemble using a CSVR thermostat with a time constant of 1 ps[92] and a MD integration time

step of 1 fs, zero field c-NNP MD simulations were carried out in the NVT ensemble using a Nose-Hoover thermostat[93,94] and a time step of 0.5 fs, except where indicated otherwise. The simulations of the orientational relaxation dynamics and the IR spectra were carried out in the NVE ensemble. The Beer-Lambert absorption coefficients were calculated as in previous work[65]. For each electric field strength (0.0514, 0.1028, and 0.2057 V Å$^{-1}$), 20 independent configurations were chosen from the field sweep PNNP simulation. For each of these configurations short (20 ps) PNNP simulations in the NVE ensemble were conducted. The spectrum was calculated for each of the 20 simulations and then averaged.

### Reporting summary

Further information on research design is available in the Nature Portfolio Reporting Summary linked to this article.

## Data availability

The source figure data are available on figshare with the following https://doi.org/10.6084/m9.figshare.26716631[95]. The supplementary data are also available on figshare with the following https://doi.org/10.6084/m9.figshare.26669350[96].

## Code availability

All field simulations were performed using the `cp2k-2023.1` software package, which has been customised in-house. This is a development version and is publicly available on: https://github.com/kjaj98/cp2k-apt-pnnp-paper, https://doi.org/10.5281/zenodo.13627849[97]. The atomic polar tensor neural network and all scripts used to train and predict atomic polar tensors are publicly available at: https://github.com/pschienbein/AtomicPolarTensor, https://doi.org/10.5281/zenodo.13323587[98]. Version 2.1.0 of `n2p2` was used to train the committee neural networks.

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

## Acknowledgements

K.J. gratefully acknowledges a PhD studentship co-sponsored by Uni-
versity College London and Pacific Northwest National Laboratory
(PNNL) through its BES Geosciences programme (FWP 56674) sup-
ported by the U.S. Department of Energy's Office of Science, Office of
Basic Energy Sciences, Chemical Sciences, Geosciences and Bios-
ciences Division. This work was further supported by an individual
postdoc grant to P.S. funded by the Deutsche Forschungsgemeinschaft
(DFG, German Research Foundation) under project number 519139248
(Walter Benjamin Programme). J.B. and P.S. acknowledge EPSRC - UKRI
for the award of computing time via an ARCHER2 Pioneer Project
(ARCHER2 PR17125). Via our membership of the UK's HEC Materials
Chemistry Consortium, which is funded by EPSRC (EP/L000202, EP/
R029431), this work used the ARCHER2 UK National Supercomputing
Service (http://www.archer2.ac.uk).

## Author contributions

K.J. implemented PNNP under the supervision of P.S. K.J. carried out and
analysed all simulations. P.S. calculated the IR spectra and helped
analyse the data. P.S. and J.B. designed the research. K.J., P.S. and J.B.
wrote the manuscript, with input from K.R. All authors reviewed and
discussed the manuscript.

## Competing interests

The authors declare no competing interests.

## Additional information

**Supplementary information** The online version contains
supplementary material available at

Philipp Schienbein or Jochen Blumberger.

**Peer review information** *Nature Communications* thanks Anders M. N.
Niklasson, Kirill Zinovjev, and the other anonymous reviewer(s) for their
contribution to the peer review of this work. A peer review file is available.

