## [Peer Review File · Nature Communications]

Machine learning the electric field response of condensed phase systems using perturbed neural network potentialsREVIEWER COMMENTS

Reviewer #1 (Remarks to the Author):

The authors have developed a surprisingly simple machine learned force field that can capture the effects of an external electric field acting on atomistic molecular systems (including condensed periodic materials). This is achieved by using a neural network that predicts the Born effective charges (or the atomic polar tensors). The paper is very well written and the results are noteworthy and presented clearly. Several interesting properties of water are investigated and carefully compared to ab initio theory, demonstrating a remarkable accuracy and fidelity. I believe the paper is very interesting and well worth publishing. However, I have a few comments that the authors may find helpful and also a few concerns about their new method that may limit its applicability that they may choose to address prior to publication.

1) From the solution of the Schrodinger equation all interatomic forces are fully determined by Coulomb interactions (or electromagnetic interactions in the relativistic case). However, often charge screening in equilibrated dense systems makes it possible to use effective short-range, charge-independent, potentials. Nevertheless, any fully general interatomic potential must include the effects of flexible charge densities and their long-range electrostatic interactions and relaxations as the atoms are moving. Here, only the interaction with an external electric field is accounted for, and the interatomic forces are all charge independent. This may lead to limitations not discussed in the article. In fact, it is somewhat of a surprise that the machine learned electric field response works so well without including any charges explicitly.

2) The authors make a strong claim that they train on zero-field reference data and that it is surprising that they still can capture the effects of finite fields. This statement is somewhat misleading, because they use the response in the interatomic forces with respect to an external electric field. That this response is calculated at zero net field doesn't matter. In fact, any analytic function (like a polynomial) can be described globally with a series expansion around its zero value. It would be better to limit this claim to their observation that it was possible to accurately capture the effects, even of strong electric fields, using

only the field-independent dipole term, i.e. that only a first-order expansion in the electric field was sufficient for the demonstrated water examples. And, as also pointed out by the authors, higher-order terms can always be included if necessary.

3) The dipoles cannot be calculated directly, only indirectly through the integration in Eq. (5). This requires a restart of the integration using ab initio data at some chosen time interval. Of course, this significantly speeds up the calculation compared to ab initio molecular dynamics simulations, but severely limits extension to large system sizes and long simulation times, at least as long as the dipole moments need to be captured.

4) The method is based on the machine learned parameterization of the Born effective charge (or APT) as in Ref. 35. or in the code at <https://github.com/pschienbein/AtomicPolarTensor>. However, the Born effective charges are not necessarily a local quantity and may, just as the atomic charges, depend on highly non-local atomic modifications (See ref 27 and the article in <https://www.nature.com/articles/s41467-020-20427-2> by Ko, Finkler, Goedecker and Behler). This limitation would therefore also limit the applicability of the proposed neural network for the electric field response.

5) The article is notably free from the usual AI-jargon, which is a relief. However, there are a number of acronyms that I feel are unnecessary, e.g. MLMD, APT etc. If there is enough space I would avoid as many acronyms as possible.

Reviewer #1 (Remarks on code availability):

I have just checked that the code exist and is openly available.

Reviewer #2 (Remarks to the Author):

In the manuscript, the authors propose an interesting PNNP approach to generate the force field of a system under an external electric field. It is based on a clear physical concept in which the perturbation energy by the applied electric field can be evaluated by dipole-field interaction. The total dipole moment can be expanded by the intrinsic dipole plus the

induced dipole up to a certain order of response, e.g. polarizability (2nd order) implemented in this work. These response properties are then expressed by atomic polar tensors learned by equivariant neural network, which add the perturbation energy to the zero-field potential learned by a conventional invariant Belter-Parrinello neural network. Given this physically inspired form, a unique feature of this approach is its ability to learn the field-induced force field with only the unperturbed zero-field training data. This approach is also further validated by calculating important dielectric properties of liquid water and the field-dependent IR spectrum of liquid water up to the field strength of ~ 0.2 V/Å with high accuracy. In this regard, PNNP is a nice and simple method that is expected to find many useful applications for systems under external electric fields. This manuscript is well organized and potentially publishable in Nature Communications. Having said that, however, there is some room for improving the current representation of the manuscript. My suggestions are listed below for the authors consideration.

First, I suggest the authors to better clarify the connection of PNNP and existing approaches on the field-dependent potential energy surface. For example, FieldSchNet also introduces the field-dependence by including the interaction between a dipole vector and field vector acting on each atom, which may be regarded as a simplified version of PNNP truncated by the first order. FieldSchNet should work in some limiting cases while fail in some other cases because of the neglect of high order interactions. Also, as mentioned by the authors in the conclusion, the current PNNP loses its reliability when the field strength becomes larger than 0.2 V/Å, i.e. even higher orders are needed to be considered. In such strong field cases, the FIREANN model should work better. A comparison with existing models will be helpful.

Figure 1 compares the RMSEs of the perturbed force and the unperturbed force. The RMSE for the unperturbed force is 90 meV/Å, while the range of the perturbed force is merely 40 meV/Å. How to reconcile these two values? In other words, the former is even larger than the absolute value of the latter, will this make the prediction of the total energy unreliable? In addition, the RMSE of 9 meV/Å at a field strength of ~ 0.2 V/Å seems too large compared to the force range (relative error is about $1/4$ of the range). This is much higher than typical force error in machine learning potential, e.g. $0.09/10 \approx 1\%$. Do I miss something here? Otherwise, I would argue the perturbation energy is less reliable at 0.2 V/Å as claimed here.

For comparison, the FIREANN model reports a RMSE for the total force of ~ 39.4 meV/Å with a more extended dataset of liquid water up to 0.6 V/Å. The authors should discuss this Figure a bit more.

In the FieldSchNet and FIREANN models, the field is acting on each atom, which can be in principle position dependent. So, these methods can simulate the dependence of the force field with an inhomogeneous electric field. It is interesting to discuss whether PNNP is also compatible with the inhomogeneous electric field that is induced for example by a STM tip or at the electrode interface.

A minor point

on page 2, when mentioning published ML models to predict dipole moments and polarizabilities, a relevant paper should not be forgot, J. Phys. Chem. B, 124, 7284, 2020, where both permanent/transition dipole moments and polarizability tensors are learned by neural networks.

Reviewer #3 (Remarks to the Author):

The authors present a combination of ML models for MLMD simulations in the presence of a uniform electric field. Using a box of 128 water molecules as a test system, they show that their approach predicts well the forces, dielectric constant and orientational relaxation compared to explicit DFT AIMD simulations.

This is a valuable contribution that aligns well with existing approaches that decouple the "internal" energy of the system from the response to the environment (for example, see 10.1063/1.5009502 for intermolecular interactions or 10.1021/acs.jctc.2c00914 for interaction with external point charges). Also, both techniques (c-NNP and APTNN) have been published previously, so the present work does not offer much of methodological novelty, but rather showcases the utility of the proposed combination of the models in practical simulations. For these reasons, this work would be a better fit for a specialist journal (for example JCTC, JCP or Commun Chem). Therefore, I would not recommend publication in Nature Communications, but fully support the authors in resubmitting the

manuscript elsewhere.

Minor comments:

1. As the authors clearly show, the E-field contribution to the total force is negligible compared to the unperturbed contribution. This explains why the model trained on zero-field data performs well - the linear response is a good approximation. However, this would probably mean that other approaches (atomic polarizabilities, Drude oscillators, fluctuating charges) would also perform very well for the case studied. A discussion of the pros and cons of their approach compared to other techniques would be useful.

2. The discussion of linear/non-linear regimes in the electric field sweep section is unclear. From the Figure 3, the orientation change seems to stay linear beyond the "linear regime" threshold all the way until the field itself starts to increase non-linearly. How did the authors decide on the threshold? Why not continue increasing the field linearly?

3. Also, a comment on how reference APTs are obtained would be helpful. If it is by finite differences, six additional SCF calculations per structure are required. It is a non-negligible overhead and should be explicitly discussed.

Reviewer comments are reproduced in black *italic*, our response to the comments are in blue and changes to the manuscript are reproduced in red.

Reviewer 1

1) From the solution of the Schrodinger equation all interatomic forces are fully determined by Coulomb interactions (or electromagnetic interactions in the relativistic case). However, often charge screening in equilibrated dense systems makes it possible to use effective short-range, charge-independent, potentials. Nevertheless, any fully general interatomic potential must include the effects of flexible charge densities and their long-range electrostatic interactions and relaxations as the atoms are moving. Here, only the interaction with an external electric field is accounted for, and the interatomic forces are all charge independent. This may lead to limitations not discussed in the article. In fact, it is somewhat of a surprise that the machine learned electric field response works so well without including any charges explicitly.

It is generally acknowledged in the field, and also shown in this work, that local descriptors work well for dipolar solvents like water where long-range interactions are effectively screened. But we do agree with the reviewer that local (i.e. 2nd generation) NNPs and local ATPNNs are bound to fail for gas-phase systems, or apolar solvents, where long-range electrostatics is no longer effectively screened and contributes much more or even dominates the energetics of the system. We have added a paragraph in main text, section "Discussion" on page 20/21: While the APT rigorously accounts for all long-range electrostatics and non-local charge transfer effects, our neural network representation of the APT (APTNN) only uses local descriptors, i.e., it does not contain explicit long-range electrostatics information. This is similar in spirit to the 2nd generation c-NNP used here to represent the interatomic potential. ML models with local descriptors are bound to fail for vapour phases or apolar media where long-range electrostatics is no longer effectively screened, or in situations where the electric field induces a long-range charge transfer. Yet, this is where the modularity of our PNNP approach offers a distinct advantage because it allows us to replace the underlying ML models as required. For example, the 2nd generation c-NNP currently modelling the interatomic potential can be replaced by a 4th generation c-NNP [72, 27, 73] which explicitly takes long-range electrostatics and non-local charge transfer into account. Similarly, the APTNN with local descriptors could eventually be replaced with a version including non-local descriptors. The idea introduced here to perturb an interatomic potential with an electric field using the APT is generally valid. It is a matter of the underlying machine learning models to incorporate non-locality when needed.

2) The authors make a strong claim that they train on zero-field reference data and that it is surprising that they still can capture the effects of finite fields. This statement is somewhat misleading, because they use the response in the interatomic forces with respect to an external electric field. That this response is calculated at zero net field doesn't matter. In fact, any analytic function (like a polynomial) can be described globally with a series expansion around its zero value. It would be better to limit this claim to their observation that it was possible to accurately capture the effects, even of strong electric fields, using only the field-independent dipole term, i.e. that only a first-order expansion in the electric

field was sufficient for the demonstrated water examples. And, as also pointed out by the authors, higher-order terms can always be included if necessary.

We agree with the reviewer that the good agreement at weak fields is less surprising. In the manuscript we thus limit our claim that it is remarkable that one can capture the effects at strong electric fields despite the learning being carried out at zero field only. In particular, we have removed the sentence in main text, page 8 “This is because the RMSE of the APT in Fig. 1D remains remarkably constant for all simulated field strengths even though it was trained exclusively on zero-field equilibrium configurations of liquid water.”

3) *The dipoles cannot be calculated directly, only indirectly through the integration in Eq. (5). This requires a restart of the integration using ab initio data at some chosen time interval. Of course, this significantly speeds up the calculation compared to ab initio molecular dynamics simulations, but severely limits extension to large system sizes and long simulation times, at least as long as the dipole moments need to be captured.*

We agree with the reviewer that this can indeed be limiting for large system sizes that cannot be routinely treated with DFT (on the order of 1000s of water molecules). We have added a paragraph in the main text Discussion on page 20: “As with any method, our current implementation of PNNP MD has some limitations which we would like to discuss in the following. First, the itinerant cell dipole moment is obtained by integrating the current density according to Eq. 5. Therefore, cell dipole moments are only indirectly obtained by numerical integration, in contrast to previously published techniques[40, 41, 43]. To compensate for the accumulation of integration errors, the integration is restarted occasionally by explicitly calculating the dipole moment by the underlying electronic structure method (Fig. 1F). While this significantly reduces the amount of electronic structure calculations required, the additional DFT calculations may become performance-limiting for very large system sizes that can no longer be routinely treated at DFT level (e.g., 1000s of water molecules). Yet, the DFT calculations do not restrict the accessible time scale that can be accessed by PNNP MD. As we show in Fig. 1F, it is sufficient to carry out DFT calculations in periodic intervals (10 ps) on the time scale accessible to PNNP MD simulations (tens of ns). Notably, these DFT calculations are only post-processing of the PNNP MD such that they can be calculated in parallel. Moreover, many relevant properties do not require knowledge of the itinerant cell dipole moment, e.g. vibrational spectra, or transport properties, where the current density (Eq. 4) is sufficient. The latter is readily available in the PNNP scheme and does not require numerical integration.”

4) The method is based on the machine learned parameterization of the Born effective charge (or APT) as in Ref. 35. or in the code at <https://github.com/pschienbein/AtomicPolarTensor>. However, the Born effective charges are not necessarily a local quantity and may, just as the atomic charges, depend on highly non-local atomic modifications (See ref 27 and the article in <https://www.nature.com/articles/s41467-020-20427-2> by Ko, Finkler, Goedecker and Behler). This limitation would therefore also limit the applicability of the proposed neural network for the electric field response.

We agree with the referee and we refer to our response to point 1 where the above article by Ko, Finkler, Goedecker and Behler (Ref. 72) and a recent article by the same authors (Ref. 73) are cited.

5) The article is notably free from the usual AI-jargon, which is a relief. However, there are a number of acronyms that I feel are unnecessary, e.g. MLMD, APT etc. If there is enough space I would avoid as many acronyms as possible.

We have removed MLMD but we retained APT as it is very often used in the manuscript. We have removed other abbreviations where possible.

Reviewer 2:

1. First, I suggest the authors to better clarify the connection of PNNP and existing approaches on the field-dependent potential energy surface. For example, FieldSchNet also introduces the field-dependence by including the interaction between a dipole vector and field vector acting on each atom, which may be regarded as a simplified version of PNNP truncated by the first order. FieldSchNet should work in some limiting cases while fail in some other cases because of the neglect of high order interactions.

We agree with the reviewer that a better comparison of our model with existing models in the literature is required. We have added 3 paragraphs comparing the introduced PNNP model with other existing models including FIREANN, FieldSchNet and SCFNN in main text, section "Discussion" on pages 17-18 including a number of additional references:

These features make our PNNP method distinct from other approaches introduced to model the interaction with external electric fields in ML simulations [39, 40, 41, 43]. The FIREANN-wF model [43] is trained on atomistic forces in the presence of an electric field to perform finite-field simulations of liquid water and thus requires field-dependent training data. The model adds a pseudo atom to each real atom which is explicitly dependent on the electric field and thereby captures the response to an applied electric field in an effective atomic dipole moment. It has been used remarkably successfully to calculate response functions, such as dipole moments and polarizabilities, and subsequently, field-dependent vibrational spectra of liquid water were computed from these quantities. In principle, this approach allows one to accurately model arbitrarily large electric fields and field gradients, as the relevant polarized configurations are explicitly included into the training set. By contrast to PNNP, in FIREANN-wF the field-induced contribution to the total energy and forces is not learned explicitly but "implicitly in the force-only training" [43]. It remains to be seen whether implicit learning is a robust strategy that can be applied to a wide range of systems.

FieldSchNet [40] incorporates external field effects by introducing vector-valued representations of atomic environments, e.g. utilizing fictitious "atomic dipole moments". It then uses vector fields to model interactions between molecules and arbitrary external environments by adding terms for dipole-field and dipole-dipole interactions. In contrast to PNNP, it requires training data that includes molecular structures in various dielectric environments and field strengths. Electrostatic response properties of the field are available as analytical derivatives of the network and have been used to compute vibrational spectra from MD simulations conducted at zero-field. Certain limitations of this method have recently been pointed out, e.g., an incomplete description of the field-system interaction when atomic dipoles are orthogonal to the applied field direction.[43]

Complementary to the above methods, SCFNN [41] relies on training the position of Wannier centers around water molecules. The method has been introduced with the main goal to incorporate long-range electrostatics, but also allows to evaluate configurations in the presence of external electric fields. The method can indeed be generalized such that molecules other than water can be treated, but "the set of possible molecules must be known in advance" [41]. Moreover, a reference frame around each kind of molecule needs to be defined; in case of a water molecule this was done by associating four Wannier centers with the molecule and examining their coordinates within the local molecular frame. This construction thus likely introduces an overhead when more molecules

need to be treated, molecular frames cannot be constructed (e.g. a simple ion) or reactions occur during the MD simulations (e.g., proton transfer in water).

Also, as mentioned by the authors in the conclusion, the current PNNP loses its reliability when the field strength becomes larger than 0.2 V/Å, i.e. even higher orders are needed to be considered. In such strong field cases, the FIREANN model should work better. A comparison with existing models will be helpful.

We note that this is the case because we observe formation of H⁺ and OH⁻ at these high field strengths and our PNNP simply has not been trained on configurations including these species. This was clarified in the main text in the original and current version, section "Results", on page 11:

"At 0.4114 V Å⁻¹, we observe water splitting into a proton and a hydroxide ion, in line with previous reports that chemical bond activation occurs at these field strengths [7]. Here one could train the c-NNP further to include proton and hydroxide species, and thereby test even larger field strengths. However, accurate description of these species would require explicit inclusion of nuclear quantum effects [53, 54], which is beyond the scope of this work. We also mention that the FIREANN-wf model was trained to work also at fields higher than 0.2 V/Å on in Discussion on p 17: "this approach allows one to accurately model arbitrarily large electric fields and field gradients, as the relevant polarized configurations are explicitly included into the training set."

2. Figure 1 compares the RMSEs of the perturbed force and the unperturbed force. The RMSE for the unperturbed force is 90 meV/Å, while the range of the perturbed force is merely 40 meV/Å. How to reconcile these two values? In other words, the former is even larger than the absolute value of the latter, will this make the prediction of the total energy unreliable?

We have added an explanation in this regard in section "Discussion" on page 19:

The excellent performance of PNNP might be surprising considering that the typical field-induced contribution to the total force, in the order of 10 meV Å⁻¹ at moderate field strengths of 0.01 V Å⁻¹ (Fig. 1C), is even smaller than the RMSEs of the unperturbed force contribution and the total force, about 90 meV Å⁻¹ (Fig. 1D). We explain this by noting that the error in the unperturbed force contribution is approximately Gaussian distributed and does not have a directional preference whereas the field-induced force contribution has, of course, a net direction along the field. Thus, when averaged over many configurations and atoms, the error in the unperturbed force contribution cancels out whereas the field-induced force contribution does not. Therefore, the RMSE in the unperturbed force contribution does not compromise the accuracy of the electric field response.

The total energy calculation is reliable as can be seen by the excellent conservation of total energy in the NVE ensemble. The energy drift is about 10⁻⁹ Hartree/ps/atom, similar as in classical MD, see Figure S1A.

In addition, the RMSE of 9 meV/Å at a field strength of ~0.2 V/Å seems too large compared to the force range (relative error is about 1/4 of the range). This is much higher than typical force error in machine learning potential, e.g. 0.09/10 ≈ 1%. Do I miss something here? Otherwise, I would argue the perturbation energy is less reliable at 0.2 V/Å as claimed here.

Please note that the force range of the E-field contribution shown in Fig 1C is for a field strength of 0.0129 V/Å (not ~0.2 V/Å). We have now added two relative error metrics,

normalising by the range and root mean square DFT forces (the latter defined in new Supplementary Note 4), an additional panel E in Figure 1 in main text and a presentation of the relative errors in section “Results” “Validation of PNNP” on page 8:

The field-induced force contributions are about two orders of magnitude smaller than the unperturbed contribution and so are their RMSEs. Notably, the field-induced force RMSE increases in proportion with the field strength from $8.6 \times 10^{-2} \text{ meV A}^{-1}$ at 0.0026 V A^{-1} to 9.05 meV A^{-1} at the largest simulated field of 0.2057 V A^{-1} . Dividing the RMSE by the range of field-induced forces one obtains 1.1% error at 0.0129 V A^{-1} and 1.2% error at 0.2057 V A^{-1} . Alternatively, dividing the RMSE by the root-mean-square of the reference DFT forces, as is commonly done in the literature[51] (see also Supplementary Note 4), we obtain a relative RMSE for the field-induced contribution of $\approx 8.2 \%$ up to a field strength of about 0.02 V A^{-1} and 9.8% at 0.2057 V A^{-1} (Fig. 1E). Hence, the relative force error on both error metrics exhibits remarkably constancy across the range of field strengths investigated.

For comparison, the FIREANN model reports a RMSE for the total force of $\sim 39.4 \text{ meV/A}$ with a more extended dataset of liquid water up to 0.6 V/A . The authors should discuss this Figure a bit more.

We have added this discussion to the present manuscript in section “Discussion” on page 19-20:

We would like to point out that the total force RMSD of our PNNP, 90 meV A^{-1} , is in line with typical values that have been previously reported for ML models, see e.g. Refs. [49, 51, 27, 50]. Yet, some recent studies reported RMSEs that are about a factor of 2-3 smaller, for instance a RMSE of 39.4 meV A^{-1} was reported for the FIREANN-wF model[43]. The training set used herein for c-NNP was not very exhaustive and consisted of only 260 configurations of a 128 water molecule box, hence it is much smaller than the training set used in Ref. [43]. Similarly, the APTNN was trained on only 27 randomly selected configurations of liquid water [38]. Another important aspect is the quality of the underlying reference DFT calculations. For consistency reasons, we employed exactly the same DFT setup that was used in previous AIMD studies on liquid water [69, 68]. In particular, we adopted a kinetic energy cutoff of 600 Ry whereas in case of the FIREANN-wF model [43] a very tight cutoff of 1200 Ry was used. It is well known that tighter convergence of the electronic structure calculations lead to less noise and smaller RMSEs in the ML model [27]. Thus, we expect that the RMSEs of our ML models could be further lowered by increasing the training set in combination with the use of tighter convergence criteria for the DFT calculations. The excellent performance of our PNNP model compared to AIMD and experimental data suggests that the current RMSE is sufficiently low for the purpose of this study.

3. In the FieldSchNet and FIREANN models, the field is acting on each atom, which can be in principle position dependent. So, these methods can simulate the dependence of the force field with an inhomogeneous electric field. It is interesting to discuss whether PNNP is also compatible with the inhomogeneous electric field that is induced for example by a STM tip or at the electrode interface.

We agree with the reviewer that this is an interesting extension of the method and we have added a paragraph to section “Discussion” on page 21: “Finally, another limitation of the current scheme is that we only considered homogeneous electric fields. For simulation under inhomogeneous fields induced, e.g., by a STM tip or an electrode interface, one would need to supplement the multipole expansion Eq. 4 with the quadrupolar interaction terms (since field gradients interact with quadrupole moments). This could be done using the Born charges obtained from the APT for the calculations of the quadrupole moment.”

4. A minor point on page 2, when mentioning published ML models to predict dipole moments and polarizabilities, a relevant paper should not be forgot, *J. Phys. Chem. B*, 124, 7284, 2020, where both permanent/transition dipole moments and polarizability tensors are learned by neural networks.

We have added the mentioned reference in the introduction.

Reviewer 3

This is a valuable contribution that aligns well with existing approaches that decouple the "internal" energy of the system from the response to the environment (for example, see 10.1063/1.5009502 for intermolecular interactions or 10.1021/acs.jctc.2c00914 for interaction with external point charges).

Our work shares with the above cited machine learning approaches the very general idea that the total energy can be broken down in individual contributions. In the first work, atom-in-molecules (AIM) properties are trained by a machine learning model and the trained parameters are then used in a classical potential which calculates energies and forces; This was primarily done to design machine learning techniques with improved transferability. The second work decouples the energy in a QM/MM fashion to implement electrostatic embedding with machine learning. Therein, an existing machine learning model is combined with a molecular mechanics force field, in particular point charges, as already mentioned by the reviewer.

We would like to emphasize that the methodological approach and the envisaged applications of our PNNP method presented here differ very substantially from the works cited above. We partition the total energy to treat the interaction with an external electric field perturbatively and we calculate the resultant forces in the rigorous framework of the atomic polar tensor. This enables us to do molecular dynamics with finite fields at ab-initio accuracy opening up novel applications in computational condensed matter physics and electrochemistry. We would like to clarify that we do not make use of AIM (first reference above) and we do not combine methods at different levels of theory as in QM/MM (second reference above) whilst neither of the two references above uses the atomic polar tensor. In PNNP the interactions of all parts of the system are treated at the same level of theory thereby avoiding any ambiguities that could arise from system partitioning when simulating condensed phase systems, e.g. aqueous solutions or electrolytes.

Both works referred to by the referee are now cited in main text (Refs. 28, 29), section "Introduction" on page 2, "Originally, these ML models were designed to calculate potential energies and forces only, including schemes that decouple the total energy of the system into internal and environmental contributions [28, 29]."

Also, both techniques (c-NNP and APTNN) have been published previously, so the present work does not offer much of methodological novelty, but rather showcases the utility of the proposed combination of the models in practical simulations. For these reasons, this work would be a better fit for a specialist journal (for example JCTC, JCP or Commun Chem). Therefore, I would not recommend publication in Nature Communications, but fully support the authors in resubmitting the manuscript elsewhere.

With due respect, we do not agree with the referee that our work merely "showcases the utility of the proposed combination of the models in practical simulations" and that it "does not offer much of methodological novelty."

Atomic polar tensor neural networks (APTNNs) were originally devised to calculate IR spectra from machine learning molecular dynamics (Ref. 38 (Ref. 36 in old version)). However, in Ref. 38 the spectra were still calculated on standard unperturbed zero-field trajectories (using committee neural

network potentials (c-NNPs)) because nuclear gradients for finite field molecular dynamics within the APT framework were not available at this point, neither in the CP2K code nor elsewhere. Herein we present a derivation of the corresponding force expressions (Eq. 2-Eq. 3), a time integration algorithm for total polarization (Eq. 5) and an efficient implementation of the polarization forces in the CP2K package (section "Methods" "Implementation of PNNP"). The latter was not trivial as it required tight synchronization of force evaluation using two networks that are hosted on different platforms.

We added the following sentence to emphasize the novelty of our work, in main text, "Introduction", on page 3: "The use of the APT and its ML representation to calculate the field-induced forces on the atoms has not been explored before, to our best knowledge, and represents the major conceptual advance of this paper."

Moreover, we would like to emphasize that our PNNP implementation has allowed us to simulate field-dependent IR spectra within the rigorous APTNN framework and much more: simulation of myriad physical processes that occur at or are induced by the presence of finite electric fields can now be carried out with APTNNs at near ab-initio accuracy but with major speed-ups compared to ab-initio MD, including dielectric response properties, ionic conductivity at strong fields and field-dependent Raman spectra. The path our novel contribution has taken is thus no different from the one of other methodological advances in the field of computational molecular science: existing models (here, APTNN) are further developed, extended and combined to enable qualitatively new types of simulations that could not be done before.

The methodological advance reported in our work and the wide-ranging applications it enables deserves dissemination to a very wide audience beyond field-specific experts. In fact, our preprint published on Archive (arXiv:2403.12319) has already been noticed and cited by leading figures in the field (e.g. B. Kozinsky (Harvard Materials intelligence research)) and has led to experimentalists contacting us in regard with the interpretation of field-dependent spectra. The experimental audience would not usually consult specialist computational journals. Also the fact the SCFNN(<https://www.nature.com/articles/s41467-022-29243-2>), FIREANN(<https://www.nature.com/articles/s41467-023-42148-y>), 4G-BPNNP (<https://www.nature.com/articles/s41467-020-20427-2>) and Cassone's paper on field-induced electrofreezing of liquid water (<https://www.nature.com/articles/s41467-024-46131-z>) all were published in Nat. Comm., suggests this topic is of interest to a wide community that would look to Nat. Comm. for further developments in the area.

Minor comments:

As the authors clearly show, the E-field contribution to the total force is negligible compared to the unperturbed contribution. This explains why the model trained on zero-field data performs well - the linear response is a good approximation. However, this would probably mean that other approaches (atomic polarizabilities, Drude oscillators, fluctuating charges) would also perform very well for the case studied.

We have added a new Supplementary Note 5, "Comparison of PNNP with unpolarizable and polarizable force fields" where this is discussed.

A discussion of the pros and cons of their approach compared to other techniques would be useful. Referee 2 had a similar request. We have added a comparison of PNNP with other relevant ML techniques including FIREANN-wf, FieldSchNet and SCFNN in main text, section "Discussion" on pages 17-18 "These features make our.....proton transfer in water)."

2. The discussion of linear/non-linear regimes in the electric field sweep section is unclear. From the Figure 3, the orientation change seems to stay linear beyond the "linear regime" threshold all the way until the field itself starts to increase non-linearly. How did the authors decide on the threshold?

We have added a more detailed explanation in the caption of Figure 3: "The vertical dashed line indicates the upper limit of the linear response regime at 0.0154 V/Angstrom. It was determined from the data shown in Fig. 4 as the threshold above which the polarization versus E-field response starts to deviate from linearity."

Why not continue increasing the field linearly?

After 1.8 ns the E-field was increased by increasing increments (i.e., no longer linearly) so one could probe the effect of very large fields on the polarization within a reasonable amount of simulation time. If the increments had stayed constant excessive simulation time would have been needed to reach very large field strengths.

3. Also, a comment on how reference APTs are obtained would be helpful. If it is by finite differences, six additional SCF calculations per structure are required. It is a non-negligible overhead and should be explicitly discussed.

We agree with the reviewer and added a brief discussion on overhead in main text, section "Methods", subsection "Training of c-NNP and APTNN" on page 24. "With regard to the APT, we use the recently published APTNN developed by one us [38] containing APTs from only 27 snapshots of an equilibrated 128 water molecule box in its training set at the level of RPBE-D3 [38]. The resultant 10,368 APTs were obtained by single point finite difference calculations. While these calculations add a non-negligible computational overhead, they can be trivially carried out in parallel on modern computing architectures. Hence, in practice the reference calculations for APTs do not represent a computational bottleneck provided that inexpensive functionals (e.g. GGAs) are used.

REVIEWER COMMENTS

Reviewer #1 (Remarks to the Author):

I was happy with the original manuscript and the additional revision have further improved the paper. I think the work is noteworthy and of significant interest to a broad audience in a rapidly evolving area of research. The methodology is sound and the results are well supported and explained. I can't see any flaws or reasons for additional revision. The software is openly available and the results should therefore be reproducible. I recommend publication without any further revision.

Reviewer #1 (Remarks on code availability):

I have only checked that the code is available and looks ok, but no test were performed.

Reviewer #2 (Remarks to the Author):

The authors have addressed my concerns properly. Now the comparison of different approaches is clearly discussed and the errors of the PNNP model are well explained. I will recommend the publication as is.

Reviewer #3 (Remarks to the Author):

Given that the other reviewers agree with publishing the manuscript in Nat. Comm., I won't insist on submitting it to another journal. The work is of high quality and has only improved with the revision.

My only remaining concern is related to the computational cost of APTs. If I understand correctly, APT for each atom requires six single-point calculations. Therefore, for the 10,368 APTs required to train the APTNN, a total of $10,368 * 6 = 62,208$ calculations were necessary. This is equivalent to 62 ps of ab initio QM MD. This represents more than a "non-negligible computational overhead" compared to regular MLP, which predicts energies and forces and would require at least an order of magnitude fewer estimates.

Instead of stating that it "can be trivially carried out in parallel on modern computing architectures", an explicit quantitative discussion of the computational effort required to train APTNN and potential ways to reduce it would be helpful. For instance, the fact that only 27 snapshots were sufficient to train the model is remarkable. Perhaps the cost could be reduced by calculating APTs for only a subset of atoms in each snapshot, such as those with the most dissimilar environments?

Reviewer 3

*My only remaining concern is related to the computational cost of APTs. If I understand correctly, APT for each atom requires six single-point calculations. Therefore, for the 10,368 APTs required to train the APTNN, a total of $10,368 * 6 = 62,208$ calculations were necessary. This is equivalent to 62 ps of ab initio QM MD. This represents more than a "non-negligible computational overhead" compared to regular MLP, which predicts energies and forces and would require at least an order of magnitude fewer estimates.*

Instead of stating that it "can be trivially carried out in parallel on modern computing architectures", an explicit quantitative discussion of the computational effort required to train APTNN and potential ways to reduce it would be helpful. For instance, the fact that only 27 snapshots were sufficient to train the model is remarkable. Perhaps the cost could be reduced by calculating APTs for only a subset of atoms in each snapshot, such as those with the most dissimilar environments?

We agree with the referee on this point. We have added an additional paragraph to the manuscript in section *Methods* on page 24 to give a quantitative discussion of the computational effort.

“The resultant 10,368 APTs were obtained by single point finite difference calculations, which required a significant computational effort. Notice that the goal of the present work was to introduce the PNNP approach and to give a proof-of-principle demonstration, not to optimize the training protocol. Here, we just used the largest training set that was already available from our previous work [38]. This has allowed us to benchmark the accuracy of PNNP whilst ensuring that possible errors from using too small training sets are minimized.

We are very confident that the computational effort for generation of training data for APTNN can be significantly reduced in the near future. First, we previously showed that the IR spectrum requires only 9 training configurations for a 128 water molecule box (i.e., 3,456 APTs) to accurately reproduce the reference spectrum, reducing the required reference calculations by a factor of three. We anticipate that this applies to other observables of interest as well. Second, our current training strategy of the APTNN has not yet been optimized. Calculating APTs for all atoms in a given configuration likely includes redundant data due to the high correlation between APTs of neighbouring atoms. Training on a per-atom basis, where APTs are calculated for a subset of atoms only, is expected to be more efficient. This method could also be combined with active learning, where the network iteratively selects atoms for the training set; for example by using a query-by-committee approach as in the c-NNP [45]. We expect that these two strategies allow us to reduce the number of reference APT calculations by an order of magnitude, if not more. Finally, the method for calculating APTs could also be made more efficient, for example, by using analytic density functional perturbation theory [88] rather than finite difference calculations. Exploring these strategies is beyond the scope of this manuscript but will be addressed in future work.”

Unrelated to the referee comment, we have continued to run (computationally expensive) ab-initio molecular dynamics (AIMD) trajectories during the reviewing process of this paper to further improve the statistics and reduce the statistical error bars for the data shown in Figure 2. We managed to obtain 10 additional AIMD trajectories during this time, increasing the total number of trajectories from 10 (as reported in the originally submitted manuscript) to 20. We are very pleased to report that due

to the improved statistics the agreement between PNNP and AIMD for the orientational relaxation dynamics is now even better than reported in the original version of this manuscript. We have updated Figure 2 and the main text on page 9/10:

20 independent configurations

$\tau = 5.9 \text{ ps}^{-1}$ ($R^2 = 0.99$) for PNNP compared to $\tau = 6.6 \text{ ps}^{-1}$ ($R^2 = 0.99$) for AIMD.

final average angle of 60° and 61° , respectively.

remaining small differences

We also added an additional detail in Simulation Details on page 26,

The simulations of the orientational relaxation dynamics and the IR spectra were carried out in the NVE ensemble.

REVIEWERS' COMMENTS

Reviewer #3 (Remarks to the Author):

The authors have properly addressed my concerns, so I am happy to recommend the manuscript for publication.